# Effect of the Absence of Unethical Controlling Shareholders on Firm Value and the Moderating Role of Corporate Governance: Evidence from South Korea

Ji-Hyun Lee and Su-Yol Lee *

College of Business Administration, Chonnam National University, Yongbong-ro 77, Buk-gu, Gwangju 61186, Korea; 186634@jnu.ac.kr
* Correspondence: leesuyol@chonnam.ac.kr; Tel.: +82-62-530-1446

**Abstract:** Prior research on unethical controlling shareholder is limited. This study examines the effect of the evanishment of unethical controlling shareholders' risk on firm value and how corporate governance moderates this effect from a principal–principal agency perspective. This research proposes a contingent model of corporate governance as a mechanism to provide professional managers with managerial autonomy. This study identifies 43 cases of controlling shareholders of Korean conglomerates being absent due to their imprisonment from 2006 to 2015. The regression analysis results indicate that the evanishment of controlling shareholders' risks does not significantly influence the affiliated firms' value. This study supports the positive effect of corporate governance on firm value. Although the statistical significance is low, it observes a tendency for corporate governance to amplify the relationship between the dissolution of unethical controlling shareholders' risks and firm value. This study contributes to the literature by being one of the first to explore unethical controlling shareholders' risks based on corporate governance theory.

**Keywords:** controlling shareholder; unethical behavior; principal–principal agency problem; imprisonment; corporate governance; autonomy effect; firm value

## 1. Introduction

Corporate governance and controlling shareholders have received increasing attention in recent decades, particularly since the occurrence of corporate scandals in the early 2000s [1]. The agency problem, which indicates conflicts of interest between managers and shareholders [2], has been addressed in the finance and accounting literature. Previous studies exploring effective corporate governance that might help mitigate the conflict rift between the management and shareholders have provided inconclusive results [3]. Furthermore, the primary agency problem has evolved to a more complicated "principal–principal agency problem" with the emergence of conflicts between large and minority shareholders [4]. Unlike Western countries where corporate ownership is dispersed, the ownership structure in most other countries, particularly in Asia and Latin America, is highly concentrated with large shareholders [5,6]. Concerns about some dominant controlling shareholders having excessive control power to extract private benefits, to the detriment of minority shareholders, have continued to grow [7,8]. Such large shareholders have an incentive to make a decision in an inappropriate and unethical way to fulfill their own interests. For instance, more than 50 owner–managers of Korean large business groups, referred to as chaebols (This study uses conglomerate, *chaebol*, and large business group interchangeably. Similarly, controlling shareholder, owner–manager, dominant shareholder, and "absolute power" are also used interchangeably.), have been involved in corporate scandals and business crimes, including embezzlement, bribery, stock price manipulation, breach of fiduciary duty, and accounting fraud. Some controlling shareholders of the

biggest Korean conglomerates, including Samsung, Hyundai, and SK Groups, have been imprisoned following their involvement in corporate scandals.

The unethical behavior of controlling shareholders, labeled as "absolute power," has caused long-standing disputes, contributing to political, social, economic, and business issues in South Korea for decades. Corporate scandals associated with business groups' controlling shareholders have been frequently reported in emerging economies such as China and Latin American countries [7–9]. Criticisms have continued to increase, stating that legal sanctions against owner–managers' unethical behavior are relatively loose [10] because of excessive concerns regarding the negative impact of their absence on the national economy. This "chaebol-overlooking" phenomenon has been more common in emerging capital markets. The clear separation between ownership and management is not well-practiced compared to the Western capital market [6]. Considering the increase in corporate risks resulting from the ultimate controlling shareholders' unethical behaviors, an inquiry into how the capital market reacts to such issues has drawn significant academic attention. For instance, on 16 February 2017, Lee Jae-yong, Vice Chairman of the Samsung Group, the largest conglomerate in South Korea, was imprisoned for embezzlement, bribery, and perjury. He was accused of bribing the former president of South Korea, Park Geun-hye, in 2015 (who resigned for her impeachment one year later). He is the "absolute leader" of the Samsung Group as its heir, largest shareholder, and chief operating officer. The stock price of Samsung Electronics Inc., one of the affiliate firms of the Samsung Group, has risen by 46.82% in 9 months since the resolution of an unethical controlling shareholder risk. This event has aroused academic curiosity about how removing dominant controlling shareholders' ' risks affects firm value in the capital market.

The purpose of this study is to investigate the effect of controlling shareholders' risks stemming from unethical behavior and business scandals on firm value and whether corporate governance attenuates or accentuates this effect. Accordingly, the current study refers to three relevant research streams: corporate governance, controlling shareholders, and corporate scandals. However, management studies on unethical controlling shareholders' effect on firm value have been limited for several reasons. First, a priori research has tested the relationship between controlling shareholders and corporate governance in various country settings [5,6,11,12]. For example, Fang et al. [12] and Nguyen et al. [5] provide evidence that controlling owner–managers might mitigate the agency problem in the Chinese and Vietnamese contexts, respectively. However, the literature on corporate governance rarely addresses the latent risks resulting from unethical controlling shareholders. Second, large shareholders and corporate executives (i.e., employed professional managers) generally play different roles in the Western capital market. Thus, owner–manager-related scandals of listed companies are unusual. This is because family-owned businesses are usually not listed companies and in the case of listed companies, unethical behavior or corporate scandals (e.g., accounting fraud) are typically related to professional managers and not shareholders. These have been long-standing issues associated with the principal–agency arrangement [13]. Third, a relevant research stream has observed the effect of corporate scandals on firm value [9,14,15]. For example, Hung et al. [9] find that corporate scandals negatively affect market credibility in China, thereby hurting financial performance. Studies on how the capital market responds to the unethical behaviors of Korean chaebol's controlling shareholders provide mixed results [14,15]. Another related stream of literature observes the absence of corporate executives; however, it focuses on accidents (e.g., hospitalization) rather than corporate scandals. The literature that explored the absence of chief executive officers (CEOs) (e.g., [16]) has rarely addressed unethical behavior and business scandals as the cause for such absence. There is a deficit in exploring the impact of the evanishment and resolution of unethical owner–managers' risks on firm value.

This study suggests a contingency model and empirically examines how the evanishment of unethical controlling shareholders' risks (due to imprisonment) affects affiliates' firm value by considering corporate governance as a moderator. We identify 43 absence

cases of the dominant owner–managers of Korean conglomerates from 2006 to 2015. We compiled a list of unethical behavior and corporate scandal cases, including embezzlement, stock manipulation, breach of duty, and illegal funds, among others, of executives who are the controlling shareholders of Korean conglomerates. Regression analysis results indicate that the evanishment of controlling shareholders' risks does not significantly influence the affiliated firms' corporate value. However, this study finds that there is a tendency for good corporate governance to moderate the relationship between controlling shareholders' unethical behavior and firm value.

This study makes three contributions to the literature. First, this is one of the foremost attempts to explore the effect of the largest shareholders' unethical behavior on firm value. The study adds to the corporate scandal literature that primarily focuses on the capital market's reaction to business crimes, scandals, and unethical behavior [9,14,17]. It extends the current literature by examining the absence and resolution of the risks resulting from the misconduct of controlling shareholders. Second, this research presents empirical evidence on how firm value changes when unethical controlling shareholders disappear, thus extending the understanding of business ethics and corporate governance. Third, the study proposes a new explanation of the relationship between the evanishment of controlling shareholders' overused power and affiliated firms' value through the principal–principal agency lens. We consider corporate governance as a moderating variable. A business group is generally under the influence of its largest controlling shareholder; however, this study confirms that affiliated companies' stock prices do not covary while the controlling shareholder is absent.

The rest of this paper is organized as follows. Section 2 presents the research framework and hypotheses based on a comprehensive literature review on the relevant research streams. Section 3 describes the research method, and Section 4 presents the results and discussion of the empirical analysis. Section 5 provides the conclusions, theoretical and practical implications, limitations and future scope of the study.

## 2. Theoretical Background and Hypotheses Development

### 2.1. Controlling Shareholder and Agency Theory

This study relates to three strands of literature: corporate governance, controlling shareholder, and corporate scandals. First, agency theory serves as a fundamental theoretical background to articulate the relationships among corporate governance, controlling shareholders, and firm value. Agency problem, first introduced by Jensen and Meckling [2], states that there might be conflicts between the interests of shareholders (i.e., principal) and employed managers (i.e., agent). Managers have the incentive and ability to maximize their own self-interest at the expense of principals. The literature on finance and accounting has explored effective and useful management control mechanisms to lessen agency costs [3]. Since the Organisation for Economic Co-operation and Development (OECD) established its first principles of corporate governance in 1999, corporate governance has continued to expand its coverage, considering ownership structure, the roles and responsibilities of shareholders, boards of directors, and the timely and accurate disclosure of information [5]. The literature has examined the characteristics of corporate governance, such as board size [18,19], board independence [20,21], management ownership [22], government ownership [23,24], and foreign ownership [25], that cause agency problems. However, the literature tends to agree that there is no single model of good corporate governance [26].

Second, the ownership structure issue generates a principal–principal agency problem as conflicts between large and minority shareholders become significant [4]. There has been a debate on the effect of concentrated ownership on agency costs. Finance literature has explored large shareholders' role in reducing agency problems between owners and managers. Advocates for controlling shareholders state that concentrated ownership is an effective monitor of executives. Collective action and free-riding problems among dispersed and minority shareholders are believed to hamper the effectiveness of supervising managers' behavior. In contrast, major shareholders have sufficient resources and incentives

to monitor and check management decisions [27]. Having such absolute power over executives can enhance their monitoring effectiveness [12]. Collectively, concentrated ownership is likely to be effective in resolving a trade-off between managerial control and interest [28]. However, an opposing view has been also receiving increasing support. The dominant shareholder has the incentive to use their controlling power in their interest by extracting private benefits, to the detriment of minority shareholders, which supports the expropriation hypothesis [4,11,29]. Previous studies depict several ways of expropriation by dominant owner–managers. They can make suboptimal investment decisions through less profitable mergers and acquisitions that may not pay off for the firm [30,31]. Furthermore, ownership concentration may hamper the adoption of effective corporate governance mechanisms [6]. Controlling shareholders with absolute power tend to substitute for the formal corporate governance system, decreasing the relevance of the board function [32].

The third strand of the literature relevant to this study observes the effect of corporate scandals on firm value. However, there exist limited related studies in finance and accounting literature. Instead, much of the literature has focused on market credibility regarding corporate scandals [33,34]. For instance, Karpoff et al. [34] provide evidence that accounting misconduct negatively affects firm value by decreasing market credibility. In emerging economies, there might be a greater chance for the occurrence of potential corporate scandals related to political ties than in Western economies. Hung et al. [9] find that corporate sandals are associated with poor stock returns in China; however, the negative effect of political scandals on firm value is more significant than that of market scandals. The literature implies that the capital market might downgrade companies when detecting risks engendered by corporate scandals and the misconduct of controlling shareholders.

### 2.2. Hypothesis Development

There have been long-standing disputes on the repeated occurrences of controlling shareholders' scandals (e.g., [5]); however, there the literature has lacked in exploring how the capital market reacts to the removal and resolution of unethical controlling shareholder risks [5,6]. Unlike the Western capital market where ownership and management are fully separated in listed companies, controlling owner–manager issues are common in most of the other countries, including Korea, China, Vietnam and Latin America [5,6,9]. This study argues that the dissolution of dominant shareholders' risks (due to imprisonment) enhances affiliated firms' value; furthermore, corporate governance positively moderates the effect.

The finance literature has examined controlling shareholders' role in mitigating agency problems between owners and managers. Collective action and free-riding problems among dispersed and minority shareholders are believed to hamper the effectiveness of supervising managers' behavior. In contrast, major shareholders have sufficient resources and incentives to monitor and check management decisions [27]. Ownership concentration creates a principal–principal agency problem. For instance, Grossman and Hart [35] are concerned that major shareholders tend to use their controlling power for their interests, which leads to disadvantages for the minority shareholders. Previous studies have provided empirical evidence on the negative effect of ownership concentration concerning controlling shareholders on firm value [35–39]. In emerging economies, ownership concentration is also likely to increase agency costs, which lowers firm value in the capital market [6,40–42]. Such phenomena are generally delineated in the managerial entrenchment hypothesis [43,44]. These results imply that the dissolution of controlling shareholders might decrease ownership concentration and agency costs, enhancing their firm value.

The principal–principal agency problem is a serious issue wherein the dominant controlling shareholders act as owner–managers. For instance, Korean chaebols have interlocking-ownership structures among subsidiaries that preserve family control despite low direct ownership [45]. In other emerging economies, such as China, Vietnam, and Brazil, a small number of controlling shareholders tend to have significant and excessive influence on affiliated firms even though these companies are publicly traded [1,5,6,46].

Fan and Wong [47] provide evidence that the controlling shareholders in Asian firms are likely to take advantage of minority shareholders by adjusting profits and limiting information usability. Dominant controlling shareholders may also manipulate information disclosure to hide their self-dealings [1,48]. A largely dispersed ownership structure effectively reduces information asymmetry problems than the ownership concentrated on the dominant shareholder [49]. The controlling shareholders in emerging economies are believed to have stronger control over affiliated firms, which implies that firm value can be more vulnerable to the controlling decision-makers. This study labels this situation as the "controlling owner–manager risk" because uncertainties regarding controlling shareholders might negatively affect the related firms. Affiliated firms under the control of such dominant shareholders might be undervalued and traded at a discount. Collectively, the evanishment of controlling owner–manager risk increases firm value [50].

This study has comprehensively reviewed empirical evidence on the effect of controlling shareholders' misconduct in Korea. Many previous studies on this topic have supported an argument that the absolute power's unethical behavior negatively affects firm value. For example, So [51] finds that some Korean companies' cumulative excess returns, whose executives had been accused of embezzlement and breach of duty, were negative. Lee and Joe [14] report that foreign investors react to corporate scandals with concern. The proportion of foreign shareholders plummeted after reports on corporate scandals such as embezzlement, dereliction of duty, illegal fundraising, and unfair trade were published. These companies' excess returns reportedly decreased shortly after these adverse events but recovered in the long run. However, such results cannot be generalized because some recent studies with a large sample of data provided mixed results. For instance, Lee and Choi [10] analyze several Korean companies whose owner–managers were convicted for embezzlement but find no significant consequence. It is noteworthy that the capital market's reactions to these owner–managers' convictions vary among affiliated companies.

In general, dominant and controlling shareholders are extremely common among large conglomerates, and they have significant and excessive control over affiliated companies, putting them at the controlling owner–manager risk. The evanishment of unethical controlling shareholders might be an opportunity to dissolve owners' uncertainty and decrease agency costs, generating a positive response from the capital market. These arguments lead to the following hypothesis.

**Hypothesis 1 (H1).** *The removal of unethical controlling shareholders due to imprisonment is positively associated with the affiliated firms' value in the capital market.*

This study proposes a contingency model to depict the effect of the controlling shareholder's evanishment by considering the moderation of corporate governance as a proxy for professional managers' discretionary decision-making power. Good corporate governance facilitates communication between shareholders and management by ensuring managerial autonomy and shareholders' involvement. It consists of various dimensions, including soundness and independence of the board of directors [52–55]. First, the board of directors should be organized to represent various shareholders' interests beyond the interest of a specific group of shareholders. The soundness of the shareholder structure indicates that corporate governance is systemized to prevent the excessive influence of a small number of controlling shareholders and reduce biased decisions [56]. Second, the independence of board activities and the fairness of the shareholders' resolution process should be secured to help guarantee their appropriate supervision and monitoring, such that distorted managerial decision-making (for the interest of managers and a limited number of shareholders) does not occur [21,26]. Collectively, good corporate governance is likely to secure general shareholders' and investors' interest by encouraging managers to utilize corporate resources more efficiently.

The literature highlights the positive effect of good corporate governance on firm value. For example, Bushman and Smith [57] and Klapper and Love [58] provide empirical evidence that effective governance contributes to corporate performance and market

value by reducing information asymmetry and balancing the interests of internal and external shareholders. Dittmar and Mahrt-Smith [59] also suggest that firms with poor corporate governance dissipate cash quickly in ways that significantly reduce operating performance. Previous studies of the Korean capital market provide consistent results that corporate governance positively affects firm value and rational decision-making about finance [42,56,60,61]. Good corporate governance systems that secure a sound board structure and independent board activities lower the excessive influence of a conglomerate's controlling shareholder over its affiliated firms. As such, the managers' autonomy and decision-making powers are not threatened. It could possibly be a form of empowerment given to professional managers. Being protected by independent boards, managers are likely to allocate corporate resources more efficiently, thereby enhancing firm performance in Korea [62–64].

The managers of such affiliated firms have the discretionary power to make appropriate managerial decisions on ordinary days. However, the controlling shareholders may have intentions of increasing their leverage on each affiliated firm, undermining corporate governance and restricting competent managers' abilities [6,32]. We argue that corporate governance may represent the discretionary power of the executive managers of affiliated firms being independent of the controlling shareholders' excessive influence. The absence of dominant controlling shareholders' power paradoxically alleviates their overusing power and reinforces the discretionary power of the employed manager, which is called the "autonomy effect" in this study.

The recent literature on digital transformation and innovation addresses how digital and technological innovation affects corporate governance (e.g., [65]). Recent digital innovations, including artificial intelligence, machine learning, and big data, have improved the visibility of management systems and reduced agency costs. For instance, the knowledge application of professional managers accentuates the positive impact of knowledge management on firm innovation, which increases managers' discretion without increasing agency costs [66,67]. Moreover, inbound open innovation and machine learning increase the discretionary power of managers, enabling them to actively invest in products, processes, and business innovation [68]. In general, digital transformation is positively associated with corporate governance and firm value [69].

Collectively, corporate governance, representing a sound and independent corporate structure, can intensify the autonomy effect of managers and thus accentuate the positive effect of the dissolution of unethical controlling shareholders' risks on firm value. These arguments lead to the following hypothesis.

**Hypothesis 2 (H2).** *Corporate governance positively moderates the relationship between the absence of unethical controlling shareholders due to imprisonment and firm value.*

## 3. Research Methodology

### 3.1. Variables and Measures

This study examined the effect of the absence of unethical controlling shareholders on their affiliated firms' value and the moderating effect of corporate governance in the South Korean context. First, this study used an independent theoretical variable, measuring the removal of unethical controlling shareholders. It employed a binary measure for this variable, that is, "the absence of controlling shareholders (i.e., owner–managers) due to imprisonment." We compiled a list of corporate scandal cases that controlling shareholders (i.e., chairman, vice chairman, president, vice president, or CEO) of Korean conglomerates were involved in; they were imprisoned for unethical behavior and corporate scandals, such as embezzlement, stock manipulation, breach of duty, and illegal funds. This method has been used in the literature on corporate scandals and business ethics [9,33,34].

Second, we used corporate governance ratings provided by independent third-party rating agencies. The Korea Economy Justice Institute's (KEJI) social rating database was the most widely used by academic researchers before 2010 to measure Korean companies'

corporate social performance (CSP). The Sustinvest Rating Database, a paid database, is currently a comprehensive and reliable CSP evaluation resource and is widely used among investment managers to determine socially responsible investment. We compiled the corporate governance rating section and standardized them for analysis.

Third, we used Tobin's Q as a proxy for firm value, which is the dependent variable. It is most widely used in the finance literature as a measure of firm value [63,70]. Fourth, we employed certain control variables. The finance literature suggests that financial leverage, operating cash flow, net income, sales growth rate, and firm size should be controlled [71,72]. We also controlled for ownership structure as measured by a fraction of the largest shareholder. This study also includes industry, year, and audit period as dummies Table 1 summarizes the variables and measurements used in this study.

**Table 1.** Measurement of variables.

| Variable | Code | Expected Correlation | Measurement |
| --- | --- | --- | --- |
| Independent | | | |
| Absence of the unethical controlling shareholder | Absence | + | 1 if the controlling shareholder was imprisoned; else 0 |
| Moderating | | | |
| Corporate governance | GOV | | |
| Soundness | G1 | + | Standardized value of KEJI and Sustinvest ratings on a sound board structure and the board's independence |
| Fairness | G2 | + | Standardized value of KEJI and Sustainvest ratings on information transparency and procedural fairness of resolution |
| Dependent | | | |
| Tobin's Q | VALUE | | (common stock market value + book value of liability)/total assets |
| Control | | | |
| Operating cash flow | $CFO_t$ | + | Operating cash flow/total assets |
| Net income | $ROA_t$ | + | Net income/total assets |
| Sales growth rate | $GRW_t$ | + | $(Sales_t—Sales_{t-1})/Sales_{t-1}$ |
| Debt ratio | $LEV_t$ | − | Total liability/Total assets |
| Firm size | SIZE | + | Log(total assets) |
| Largest shareholder's holdings | OWNER | − | The percentage share of the largest shareholder's holdings |
| Audit period | FISCAL | / | 1 if the audit is in December; else 0 |

### 3.2. Sample and Dataset

This study compiled a comprehensive list of corporate scandals and business crimes committed by controlling shareholders (i.e., owner–managers) of Korean business groups who were imprisoned from 2000 to 2015. This research focused on companies listed on the Korea Stock Exchange (i.e., KOSPI and KOSDAQ). This study defined the dominant controlling shareholder as an executive (i.e., chairman, vice chairman, president, vice president, or CEO) who is also simultaneously a member of the controlling shareholder household. Following prior studies [5,6], this study identifies the dominant controlling shareholders engaged in misconduct using regulatory enforcement actions in Korea from 2005 to 2015. Misconduct and unethical behavior include "business leader detention," "embezzlement," "stock price manipulation," "breach of duty," and "public criticism over businessperson."

This study found 52 owner–managers of Korean chaebols involved in unethical corporate scandals and correspondingly compiled 92 cases of imprisonment from 2000 to 2015. Embezzlement was the most frequent cause for their imprisonment accounting for 22.4% of the cases under study, followed by breach of fiduciary duty (16.4%) and accounting fraud (15.5%). Moreover, controlling shareholders' misconduct was found in various areas of social concerns, including mortgage fraud, slush fundraising, tax evasion, political bribery, stock manipulation, and personal scandals such as gambling and assault. Among such

cases, 44.6% resulted in imprisonment while the leaders were working as executives and 55.4% resulted in confinement after the leaders stepped down from their respective positions. This indicates that more than half of the cases of unethical scandals were detected and led to indictments much later than when the actual crimes were committed. Twelve controlling shareholders were imprisoned at least two times, which implies that unethical corporate scandals are likely to occur repeatedly.

This study used 43 imprisonment cases of controlling shareholders of listed Korean firms from 2006 to 2015 because data on corporate governance have been available only since 2006. This research used datasets of good corporate governance ratings obtained from KEJI (2006 to 2009) and Sustinvest (2010 to 2015). All scores were standardized on a yearly basis to reduce the possible biases arising from rating databases and years. The financial data used in this study were drawn from the National Information & Credit Evaluation Inc.'s KIS-VALUE dataset. A total of 2497 Korean firms were considered for the analysis.

### 3.3. Empirical Model

To test the hypotheses, this study employed the following econometric models:

$$VALUE_{i,t} = \beta_0 + \beta_1 Absence_{i,t} + \beta_2 ROA_{i,t} + \beta_3 GRW_{i,t} + \beta_4 LEV_{i,t} + \beta_5 CFO_{i,t} + \beta_6 SIZE_{i,t} + \beta_7 OWNER_{i,t} + \beta_8 FISCAL_{i,t} + \sum IND + \sum YEAR + \varepsilon \qquad (1)$$

$$VALUE_{i,t} = \beta_0 + \beta_1 Absence_{i,t} + \beta_2 ROA_{i,t} + \beta_3 GRW_{i,t} + \beta_4 LEV_{i,t} + \beta_5 CFO_{i,t} + \beta_6 OWNER_{i,t} + \beta_7 FISCAL_{i,t} + \beta_8 SIZE_{i,t} + \beta_9 GOV_{i,t} + \beta_{10} GOV_{i,t} \times Absence_{i,t} + \sum IND + \sum YEAR + \varepsilon \qquad (2)$$

First, this study examined the effect of the absence of unethical controlling shareholders on affiliated firm value using the pooled ordinary least square regression. Equation (1) was used to test Hypothesis 1. Second, this research used a hierarchical regression analysis by adding corporate governance to analyze its moderating effect (Equation (2)). The Hurber–White covariance method was employed to minimize the possibility of a heteroskedasticity problem, which is likely to happen when there is a huge gap of samples between a group of interest (i.e., "evanishment-affiliated" companies of the imprisoned controlling shareholder) and a group of comparison (i.e., other listed companies).

## 4. Results and Discussion

### 4.1. Descriptive Statistics and Correlations

Table 2 represents the descriptive statistics and correlations for the variables used in this study. The dependent variable, Tobin's Q, is not significantly correlated with the evanishment of unethical controlling shareholders. It is noteworthy that corporate governance, which is categorized into the two variables soundness (G1) and fairness (G2), has a positive and significant correlation with firm value (i.e., Tobin's Q) at the 0.01 cut-off level. Moreover, Tobin's Q appears to be highly correlated with the key control variables, presenting positive correlations with profits (ROA), sales rate (GRW), operating cash flow (CFO), firm size (SIZE), and ownership concentration (OWNER).

**Table 2.** Descriptive statistics and correlation analysis.

| Variable | Mean | S.D. | 1 | 2 | 3 | 4 | 5 | 6 | 7 | 8 | 9 |
|---|---|---|---|---|---|---|---|---|---|---|---|
| 1. Firm value | | | | | | | | | | | |
| 2. Absence | 0.0166 | 0.1279 | 0.011 | | | | | | | | |
| 3. ROA | 0.0367 | 0.1009 | 0.151 *** | 0.002 | | | | | | | |
| 4. GRW | 0.0760 | 0.4527 | 0.082 *** | −0.014 | 0.053 *** | | | | | | |
| 5. LEV | 1.0837 | 1.7961 | −0.020 | −0.004 | −0.274 *** | −0.019 | | | | | |
| 6. CFO | 0.0589 | 0.0805 | 0.268 *** | 0.018 | 0.369 *** | −0.026 | −0.168 *** | | | | |
| 7. SIZE | 27.1730 | 2.0105 | 0.103 *** | 0.004 | 0.020 | −0.017 | 0.179 *** | 0.073 *** | | | |
| 8. OWNER | 0.2851 | 0.1557 | 0.044 ** | 0.017 | 0.012 | −0.009 | −0.001 | 0.043 ** | 0.053 *** | | |
| 9. GOV(G1) | −2.7065 | 4.7011 | 0.147 *** | 0.026 | 0.052 *** | −0.111 *** | 0.019 | 0.086 *** | 0.218 *** | −0.019 | |
| 10. GOV(G2) | −2.7799 | 4.8409 | 0.133 *** | 0.052 *** | 0.040 ** | −0.111 *** | 0.035 * | 0.074 *** | 0.231 *** | −0.002 | 0.962 *** |

*, **, and *** denote $p < 0.1$, $p < 0.05$, and $p < 0.01$, respectively.

### 4.2. Effect of the Absence of Unethical Controlling Shareholders on Firm Value

Using the T-test, this study analyzed the differences in firm values between the absence cases and comparison groups. The absence group consists of affiliated firms whose

controlling shareholders have been imprisoned for being involved in corporate scandals. Table 3 shows the test results.

**Table 3.** Results of the T-test.

| Variables | Absence = 1 | Absence = 0 | Difference | *p*-Value |
|---|---|---|---|---|
| VALUE | 1.266 | 1.213 | 0.053 | 0.650 |
| ROA | 0.037 | 0.036 | 0.001 | 0.936 |
| GRW | 0.027 | 0.077 | −0.050 | 0.063 * |
| LEV | 1.031 | 1.087 | −0.056 | 0.510 |
| CFO | 0.069 | 0.059 | 0.010 | 0.404 |
| SIZE | 27.319 | 27.168 | 0.151 | 0.545 |
| OWNER | 0.301 | 0.285 | 0.016 | 0.518 |
| GOV: G1 | −1.900 | −2.710 | 0.810 | 0.158 |
| GOV: G2 | −0.989 | −2.801 | 1.811 | 0.003 *** |

* and *** denote $p < 0.1$ and $p < 0.01$, respectively.

The results indicate that generally, there exist no significant differences between the evanishment and comparison groups, which is inconsistent with our first hypothesis. Sales growth is higher in the comparison group than in the absence group at the 0.1 cut-off level. Corporate governance ratings are surprisingly higher in the absence group, indicating that the soundness and fairness of board structure and activities are greater than the comparison group. The firms belonging to the absence group were affiliated firms of the ten largest business groups in Korea. As such, their governance systems were relatively well-established, which resulted in the companies earning higher scores from ESG (environment, society, and governance) rating agencies.

This study further analyzed the data using regression analysis. Table 4 presents the results of the test of Hypothesis 1, which predicts a positive relationship between the removal of the unethical controlling shareholder and firm value. The regression equation is statistically significant with 17% explanatory power. However, this hypothesis is not supported in this analysis, indicating that no positive or negative relationship exists between the absence of the unethical controlling shareholder (due to imprisonment) and Tobin's Q of the affiliated firms. Although the first prediction of this study was not confirmed, the result is in line with a study by Lee and Choi [10] that find no significant relationship between the conviction of executive managers and abnormal returns in the capital market. Regarding the control variables, operating cash flow and firm size are positively associated with Tobin's Q, consistent with previous studies [73]. Collectively, the removal of the unethical absolute power of Korean conglomerates is not likely to influence the firm value of affiliated companies.

**Table 4.** Results of the regression analysis.

| | (1) VALUE$_t$ | Expected Sign |
|---|---|---|
| Absence | 0.058 | (+) |
| ROA | 0.435 | (+) |
| GRW | 0.163 * | (+) |
| LEV | 0.009 | (−) |
| CFO | 2.252 *** | (+) |
| SIZE | 0.024 *** | (+) |
| OWNER | 0.051 | (+) |
| YEAR | include | |
| INDUSTRY | Include | |
| FISCAL | include | |
| Adj. R$^2$ | 0.170 | |
| F-statistic | 22.109 *** | |
| No. of cases | 2497 | |

* and *** denote $p < 0.1$ and $p < 0.01$, respectively.

### 4.3. The Moderating Role of Corporate Governance

Hypothesis 2 postulates that good corporate governance accentuates the positive effect of the unethical controlling shareholder's removal on firm value. The results of the hierarchical regression analysis are presented in Table 5. In the analysis, the average of the two governance variables (i.e., soundness and fairness) for good corporate governance measure (GOV) was computed. The results show no significant interaction between the evanishment of the controlling shareholder and corporate governance, which does not support our second hypothesis. No direct relationship between absence and Tobin's Q was found. It is noteworthy that good corporate governance is positively associated with firm value. This finding implies that the capital market favors companies that have sound and fair governance systems. This finding is consistent with many previous studies [42,56–58].

**Table 5.** Results of the hierarchical regression analysis.

|  | VALUE$_t$ | VALUE$_t$ |
|---|---|---|
| Absence | 0.062 | 0.086 |
| GOV | 0.007 ** | 0.007 ** |
| GOV * Absence |  | 0.014 |
| ROA | 0.414 | 0.414 |
| GRW | 0.167 * | 0.167 * |
| LEV | 0.010 | 0.010 |
| CFO | 2.224 *** | 2.223 *** |
| SIZE | 0.022 *** | 0.022 *** |
| OWNER | 0.066 | 0.066 |
| YEAR | include | Include |
| INDUSTRY | Include | Include |
| FISCAL | include | Include |
| Ad R-sq. | 0.1719 | 0.180 |
| F-statistic | 21.409 *** | 20.586 *** |
| N | 2497 | 2497 |

*, **, and *** denote $p < 0.1$, $p < 0.05$, and $p < 0.01$, respectively.

Although the moderation effect of corporate governance is not supported at a significant level, the results indicate that there exists a slight possibility of corporate governance strengthening the relationship between the removal of the unethical controlling shareholder and firm value. To further investigate this possibility, this study calculated the regression equations observing the relationship between absence and firm value at high and low levels of corporate governance. This research defines high and low values as plus and minus one standard deviation from the mean [74]. Figure 1 illustrates the possible effect of moderation. Here, a high level of corporate governance is shown to reinforce the effect of absence slightly but positively on Tobin's Q, supported by a significant simple slope calculation ($\beta = 0.10$, $p < 0.05$). Conversely, a low level of corporate governance has a positive but relatively lesser impact on the relationship between the absence and firm value ($\beta = 0.07$, $p < 0.15$). This study provides strong evidence that corporate governance acts as one of the most critical factors in enhancing firm value in the South Korean capital market. Companies with sound and fair governance systems are much less vulnerable to the uncertainty posed by the controlling shareholders, which is generally understood as the chaebol problem in South Korea. Moreover, in the removal of unethical controlling shareholders due to their imprisonment, good corporate governance may enhance firm value by allowing for the discretionary power of professional managers of affiliated companies. This moderating effect of corporate governance implies the likelihood of an autonomy effect scenario in this study.

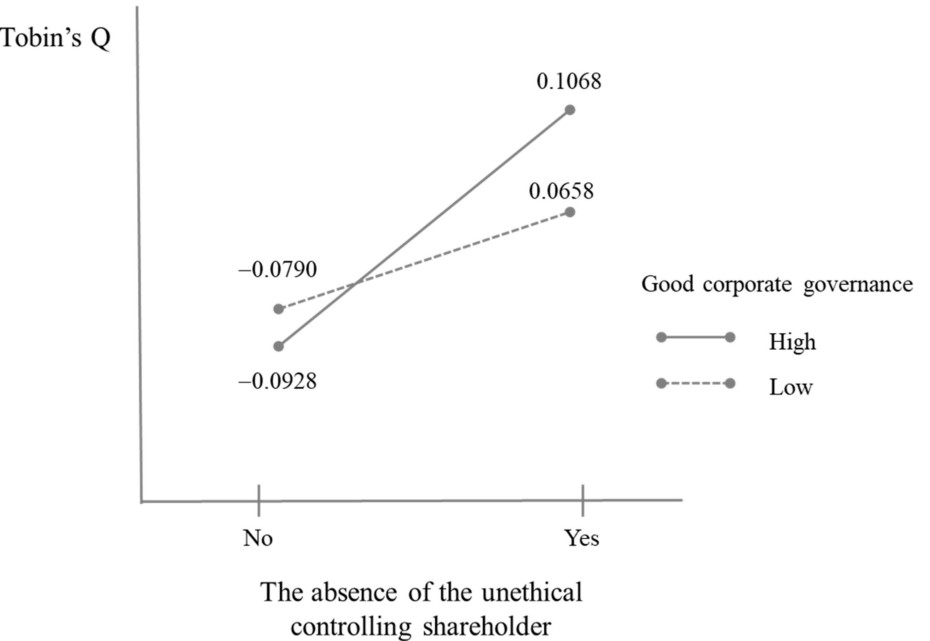

**Figure 1.** Relationship between the removal of the unethical controlling shareholder and firm value in relation to corporate governance.

*4.4. Robustness Test*

This study reports the results of an additional test conducted to determine whether our main findings are robust. This research focuses on measuring firm value at different points in time. This study regressed controlling shareholders' absence on time-lagged Tobin's Q (i.e., VALUE$_{t+1}$ and VALUE$_{t+2}$). Table 6 presents the results, which are consistent with the previous results.

**Table 6.** Robustness test with a time lag.

|  | VALUE$_{t+1}$ | VALUE$_{t+1}$ | VALUE$_{t+1}$ |
|---|---|---|---|
| Absence | 0.167 | 0.170 | 0.195 |
| GOV |  | 0.007 ** | 0.006 ** |
| GOV * Absence |  |  | 0.014 |
| ROA | 0.513 | 0.4940 | 0.495 |
| GRW | 0.090 * | 0.0941 ** | 0.094 * |
| LEV | 0.022 *** | 0.0221 *** | 0.022 *** |
| CFO | 2.161 *** | 2.1364 *** | 2.135 *** |
| SIZE | 0.017 ** | 0.0149 ** | 0.015 ** |
| OWNER | 0.027 | 0.0400 | 0.040 |
| YEAR | include | include | include |
| INDUSTRY | include | include | include |
| FISCAL | include | include | include |
| Adj. R$^2$ | 0.176 | 0.177 | 0.177 |
| F-statistic | 22.924 *** | 22.161 *** | 21.308 *** |
| N | 2497 | 2497 | 2497 |

*, **, and *** denote $p < 0.1$, $p < 0.05$, and $p < 0.01$, respectively.

First, the removal of unethical controlling shareholders is not shown to influence time-lagged firm value. This result confirms that the firm value of chaebol companies (i.e., affiliated companies) is not positively affected by the controlling shareholders' imprisonment. Second, corporate governance is positively associated with firm value at periods t+1 and t+2 at the 0.05 cut-off level. However, its moderating role is not supported in this time-lagged analysis. Third, the effects of the key control variables on a time-lagged firm

value are also similar to previous analysis results in this study. One different result is found while observing the effect of debt ratio (i.e., leverage) on Tobin's Q, which is likely to start its influence one year later. Collectively, the analysis with a time lag confirms the robustness of the previous analysis, which does not find evidence to support our hypotheses.

## 5. Discussion of Findings

### 5.1. Theoretical Contributions

This study makes three contributions to the literature. First, it is one of the first studies to explore the principal–principal agency problem from the perspective of unethical controlling owner–manager risk. Previous studies focus on the capital market's reaction to corporate scandals and misconduct issues [9,14,17]. This study adds to the finance, accounting, and business ethics literature by incorporating several critical issues related to business crimes, unethical behavior, dominant controlling shareholders, the absence of absolute power due to imprisonment, and firm value in the capital market into a comprehensive research framework. Second, it extends the body of knowledge in the literature on corporate governance and corporate social responsibility, by providing an empirical observation on how corporate governance moderates the relationship between the evanishment of controlling shareholder risks and firm value. This study confirms that affiliated firms' stock returns do not covary when unethical controlling shareholders disappear. This research presents a contingency model to better understand corporate governance as a mechanism to improve the autonomy effect of professional managers by protecting them from the excessive control of the dominant controlling shareholders. Third, this study contributes to a growing literature in international corporate finance. Most emerging economies, including China, Vietnam, and Brazil, have similar issues regarding controlling shareholders and the principal–principal agency problem [5–9]. This research helps scholars better understand the dynamic nature of unethical controlling shareholders' risks in emerging economies.

### 5.2. Practical Contributions

This study also contributes to a long-standing debate on the relationships among legal, ethics, and economic systems. Influential economic organizations such as conglomerates in Korea, Vietnam, Brazil, and other emerging economies may affect judicial decisions on dominant controlling shareholders' unethical behavior [10,17]. There has been an argument that legal and economic systems can evolve through various interactions and dialectic relationships [75]. For example, criticisms have continued to increase, stating that legal sanctions against chaebols' business crimes in Korea are relatively loose [17] because of excessive concern about the negative impact of their absence on the national economy. This chaebol-overlooking phenomenon in Korea has generated inappropriate signals in that dominant controlling shareholders may receive jurisdictional preferential treatments. However, this study implies that business risks related to unethical controlling shareholders, particularly in countries where concentrated ownership is common, should be properly addressed. Unethical behavior in business should not go unpunished. Moreover, the removal of controlling shareholders' absolute power due to imprisonment does not affect the economy.

### 5.3. Limitations and Further Research Directions

By indicating some limitations, this study suggests the directions for future research. It focuses on the evanishment of unethical controlling shareholders due to imprisonment. That is a specific situation and, thus, limits the sample. To address this limitation, there are three possible directions for future research. First, future research could consider other types of management vacancies not attributable to business crimes, such as accidents. Second, further studies could increase the sample size by considering various scandals and the misconduct of controlling shareholders, not limited to imprisonment cases [9]. Third, this study could be extended to other countries, particularly emerging economies, in which

the principal–principal agency problem is significant. Such future studies should examine differences and similarities in the absence of controlling shareholders among emerging economies. Moreover, this study used Tobin's Q as a proxy for firm value. Further studies need to re-test our research model using alternative measures for firm value, such as profits and cumulative abnormal returns, by employing the event study method. Further, it is noteworthy that the capital market might expect that imprisoned business leaders would soon return to work in the management [10,17]. Future research could compare firm value before and after the evanishment of unethical controlling shareholders and upon their returns.

## 6. Conclusions

Dominant controlling shareholders, who have tremendous and excessive influence over their affiliated firms, have been the center of long-standing disputes in political, social, and economic issues, particularly in cases where the roles of ownership and management are not clearly distinguished. The principal–principal agency problem, which has resulted in conflicts between controlling and minority shareholders has received increasing attention as corporate scandals and the misconduct of owner–managers with absolute power occur repeatedly. Dominant controlling shareholders have strong control over affiliated firms and the "controlling owner–manager risk" may force them to be more vulnerable to uncertainties. This study examined how the capital market responds when risks related to unethical controlling shareholders are resolved. It proposed the hypotheses that the dissolution of dominant shareholders' risks due to imprisonment enhances affiliated firms' value and corporate governance attenuates the relationship. This study proposed a moderating role of corporate governance as a mechanism to provide professional managers with discretionary power and managerial autonomy. The results of empirical tests, however, did not provide evidence supporting the hypotheses. The findings of this study indicate that the evanishment of unethical controlling shareholders due to imprisonment does not significantly influence firm value in the capital market. In line with previous studies, this study provides evidence that corporate governance is positively associated with firm value. Although the statistical significance is low, this study observes a tendency for corporate governance to amplify the relationship between the dissolution of unethical controlling shareholders' risks and firm value.

**Author Contributions:** J.-H.L. designed the study, collected data and conducted statistical analysis. S.-Y.L. developed a research framework and hypotheses and wrote the manuscript. All authors have read and agreed to the published version of the manuscript.

**Funding:** This research received no external funding.

**Institutional Review Board Statement:** Not applicable.

**Informed Consent Statement:** Not applicable.

**Data Availability Statement:** The data presented in this study are available on request from the corresponding author.

**Conflicts of Interest:** The authors declare no conflict of interest.

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
