# Peer review of "Effect of the Absence of Unethical Controlling Shareholders on Firm Value and the Moderating Role of Corporate Governance: Evidence from South Korea"

_sustainability, doi:10.3390/su14063607_

Round 1

Reviewer 1 Report

The study is quite interesting, and the outcomes of this paper are a valuable addition to the literature. 

This study examines the effect of the evanishment of Korean conglomerates' unethical controlling shareholders due to imprisonment on firm value by considering corporate governance as a moderator.Overall, this article is not written clearly, but it should not be granted publication in Sustainability. There are problems with English, i.e., Punctuations, grammar, and propositions. The paper needs copy editing. In addition, critical literature and discussion of findings are major and significant concerns of this article, and author(s) used the personal pronoun ‘we’ or ‘our’ throughout the paper and should replace it with the word “this study or this research” See comments below for details.

-There is a need to revise the paper title, and the current title does not portray the actual meaning of the paper. 

Abstract: The abstract is not well written. There is a need to revise with explicit contents of the abstract, i.e., the main issue, sampling, a statistical tool, methods, results, and implication. The author(s) should provide a precise and focused abstract.

  • The sampling criteria, population, unit of analysis for selecting respondents are missing. The author should highlight sampling criteria for more clarity to readers.
  • As a suggestion for improvement, the author(s) should not use the same Keywords as like Paper Title. It is encouraged to use different keywords which are not in the Paper title. It will enhance paper searchability after publication.

Introduction: The introduction section is not well written. There are ambiguous statements and no clarity in the introduction section.

  • The introduction section does not start with a broader area and issue or global context. Therefore, there is no synthesis in writing an introduction section.
  • There is less debate on the targeted topic and problems. Therefore, it will be a more valuable addition to the paper if the author(s) explain some statistics figures and recent issues.

The paper did not incorporate major literature on Corporate Governance, but the paper does not sufficiently cover recent research in the area. Helpful in this regard would include relevant research recently covered in top journals of similar scope. Further, work needs to be done to support the findings based on the current literature, as a recent theory in the area is directly counter to what was found.

  • Author(s) used underpinning theory to justify this research, i.e., agency theory etc. Although this is the major concern of this paper, the author(s) should highlight how this research contributes to the theory or contribution of the theory. Unfortunately, the author(s) did not discuss much using developed theories in this area.
  • There is a need to add more critical recent literature based on theoretical argumentation.
  • In the Introduction section, the brief discussions of methods, tools, sampling, and findings are missing.
  • An important question to answer is, “Why should Sustainability readers be interested in the results of this paper, which scrutinized only one country's data?” The reason given is not supportive. Are the findings generalizable to other developing countries like Thailand and Indonesia? The author(s) needs to improve the underwritten motivation.
  • The hypotheses development is poorly written; author(s) should cite previous studies relevant to proposed hypotheses, i.e., international and local perspectives studies in the light of underpinning theory.

Methodology: The author(s) did not develop its argument from appropriate theory and explored models previously studied in the same area.

  • What are the criteria for the selection of firms for data collection? (sampling techniques, i.e., random, cluster, or judgmental sampling)?
  • Regarding the methodology, more details and justification of why panel data analysis is not used?
  • The author(s) did not define the data collection and sampling clearly.
  • The author(s) should provide complete statistical analysis, i.e., skewness and kurtosis.
  • There is a need for improvement in reporting results such as; the author(s) should report [Beta value OR standard error (S.E) with significance level OR t-value; i.e., (β= xxx. P<0.01) OR (S. E= xxx, t > xxx). However, it could be more effective if the author(s) presents significant results with bold and asterisk (*).

Discussion and findings: As results are clearly provided. However, there is no solid discussion on results. 

  • The author(s) should discuss the limitations of this study and future research direction in a constructive way. Hence, author(s)  should write in prices and in a constructive way under a subsection of discussion.
  • The author(s) did not discuss the theoretical and practical contribution of this study. Therefore, the author(s) should discuss this study's theoretical and practical contribution in the separate subsection under discussion for more clarity.

Citation and End References

The in-text citations and end list of references do not sufficiently correspond. Please cross-check and correct citations and references throughout the paper. 

Quality of Communication: The paper needs further proofreading. I have tried to read the paper constructively, but I felt it suffers from poor writing. I, therefore, request the author(s) to pass the manuscript for professional proofreading. I suggest that a more careful investigation of prior literature can make this paper distinguishable. Linking this article with prior studies does not seem sufficient, which weakens the justification of incremental contributions.

Author Response

Please attached, our response in detail to your helpful comments and suggestions.

We really appreciate that the Reviewer is interested in our paper and giving us a chance to improve our original manuscript. We have tried our utmost to revise the manuscript considering your helpful comments, which have helped improve the quality of this manuscript. 

Reviewer 2 Report

Thank you for great pleasure reading the article! Enjoyed every page of it:-)

Suggestions:

  1. I would doubt about using "business leader imprisonment" as a key work. Maybe "controlling shareholder" would be a better option?
  2. Maybe as one of the main parts of the article "Results and discussion" should not end by the "table"... That's some style  element that would improve the article when the part ends by some textual elements.

Author Response

(The authors gave the same response as above.)

Reviewer 3 Report

This is an interesting paper and I enjoyed reading it. However, there are essential weaknesses that need to be addressed.

0) Abstract: Authors should state their contribution in terms of issue problems solved or ameliorated, theory or policy dilemmas resolved, or the like. Abstract should offer at least one example of a theoretical or managerial implication that authors concluded after their work.

1) The introductory/opening section should communicate a little clearer the literature gaps, as well as the study's aims & objectives in order to facilitate the flow of the study.

2) Overall there are good arguments and well researched points made in this paper, but I feel that author needs to take ´Theoretical development and research hypothesis´ (page 2), and ´Conclusion and discussion´ (page 10) to a further level.  The paper is interesting, but there is a lack of development in the theoretical argument and its link with the hypotheses.

It is important to read and cite (where appropriate) current literature, providing a substantial number of citations to support your work. It is also important to read (and, if relevant, cite) papers that have already been published in the JOURNAL Sustainability This will help to show the consistency of your research with the debate taking place in the journal.

The author(s) need to invest more effort in developing the linkage between recent theory and hypotheses.

Additional references to recent & relevant empirical studies could increase the quality of the research paper and provide a much clearer message to the reader - these may help you building your discussion which needs to be extended. Add the following to your reference list:

Medase, S. K. (2020). Product innovation and employees’ slack time. The moderating role of firm age & size. Journal of Innovation & Knowledge, 5(3), 151-174. https://doi.org/10.1016/j.jik.2019.11.001

Moretti, F., & Biancardi, D. (2020). Inbound open innovation and firm performance. Journal of Innovation & Knowledge, 5(1), 1–19. https://doi.org/10.1016/j.jik.2018.03.001

Ode, E., & Ayavoo, R. (2020). The mediating role of knowledge application in the relationship between knowledge management practices and firm innovation. Journal of Innovation & Knowledge, 5(3), 210-218. https://doi.org/10.1016/j.jik.2019.08.002

Haftor, D. M., Costa Climent, R., & Lundström, J. E.. (2021). How machine learning activates data network effects in business models: Theory advancement through an industrial case of promoting ecological sustainability. Journal of Business Research131, 196–205. https://doi.org/10.1016/j.jbusres.2021.04.015

Some of the statements you make are entirely obvious and should be supported in the text by these specific references.  

2) The question could be asked of whether this study is representative of other sectors in your country or in the world. Please explain this potential applicability to a general context.

3) The statistical treatment is acceptable.

4) At the end of the ´Conclusion´ section, the author should include clear statements as to where research should now go – what are the issues requiring further research and investigation? The author has to suggest challenges and possible new directions for future work. Perhaps: if the results obtained are only studied in the short term, which is then an important bias in analysing the influence further than three years in time and in their influence in the future.  

4) Concluding remarks – authors must elaborate more on what is their contribution to the literature as well as on opportunities for future research. Questions that need to be answered: Why your study is important? and how it extend so existing knowledge on the issue/topic? Conclusions need to be written in a clear and coherent manner and draw the main lessons from the paper. I suggest you to concentrate on the description of the implications of the work, the main findings and its potential replicability - empirical investigation. Furthermore, limitations of the study need to be outlined to a greater extent, and so are any potential connections between your study and specific aspects of the Journal's scope.

5) Carefully check the references, so as to make sure they are all complete and follow the Guidelines to Authors.

6) Finally, when you submit the corrected version, please do check thoroughly, in order to avoid grammar, syntax or structure/presentation flaws. Make sure you retain a formal/academic-specific style of presenting your work throughout the text - (if necessary) please seek for professional English proofreading services or ask a native English-speaking colleague of yours in order to refine and improve the English in your paper.

6) The paper needs to be revised by an English native speaker. Some expressions need to be revised and given a fresh approach by an experienced native proofreader.

Thank you for the opportunity to read the paper.

Author Response

Please find attached, our response in detail to your helpful comments and suggestions.

We really appreciate that the Reviewer is interested in our paper and giving us a chance to improve our original manuscript. We have tried our utmost to revise the manuscript considering your helpful comments, which have helped improve the quality of this manuscript. 

Round 2

Reviewer 1 Report

Dear Authors, 

Thank you for incorporating the said changes. I found that you did not develop a separate section of "Discussion of Findings".

Please note that  you must follow the following sequence

5- Discussion of Findings

Under this section, you must debate on your results and incorporate philosophical thoughts to discuss your results

5.1 Theoretical contributions 

5.2 Practical contribution

5.3 Limitations and further research directions

6- Conclusion

write your concluding arguments for this paper.

  • There is a 21% similarity index, kindly reduce below 15%.

Author Response

We appreciate that you are interested in this study and give us an opportunity to revise the manuscript one more time. In consideration of your constructive suggestions, we have revised the manuscript as follows:

  1. As you suggested, we presented the Discussion of findings section with three separated sub-sections (5.1., 5.2., and 5.3.) and the conclusion section. 
  2. We have checked the manuscript and tried utmost to reduce the similarity. We will check one more time when we receive the similarity report with the final revision.

Reviewer 3 Report

I consider it improved

Author Response

We are grateful for your interest in our manuscript. Your helpful and constructive suggestions have helped improve the quality of the manuscript.